# Endothelial Semaphorin 3fb regulates Vegf pathway-mediated angiogenic sprouting

**Charlene Watterston**[1,2], **Rami Halabi**[2,3], **Sarah McFarlane**[2,3,4], **Sarah J. Childs**[1,2]*

**1** Department of Biochemistry and Molecular Biology, University of Calgary, Calgary, Canada, **2** Alberta Children's Hospital Research Institute, University of Calgary, Calgary, Canada, **3** Hotchkiss Brain Institute, University of Calgary, Calgary, Canada, **4** Department of Cell Biology and Anatomy, University of Calgary, Calgary, Canada

* schilds@ucalgary.ca

**Data Availability Statement:** All relevant data are within the manuscript and its Supporting Information files.

**Funding:** CW was funded by a University of Calgary Eyes High Doctoral Studentship (https://

## Abstract

Vessel growth integrates diverse extrinsic signals with intrinsic signaling cascades to coordinate cell migration and sprouting morphogenesis. The pro-angiogenic effects of Vascular Endothelial Growth Factor (VEGF) are carefully controlled during sprouting to generate an efficiently patterned vascular network. We identify crosstalk between VEGF signaling and that of the secreted ligand Semaphorin 3fb (Sema3fb), one of two zebrafish paralogs of mammalian Sema3F. The *sema3fb* gene is expressed by endothelial cells in actively sprouting vessels. Loss of *sema3fb* results in abnormally wide and stunted intersegmental vessel artery sprouts. Although the sprouts initiate at the correct developmental time, they have a reduced migration speed. These sprouts have persistent filopodia and abnormally spaced nuclei suggesting dysregulated control of actin assembly. *sema3fb* mutants show simultaneously higher expression of pro-angiogenic (*VEGF receptor 2* (*vegfr2*) and *delta-like 4* (*dll4*)) and anti-angiogenic (soluble *VEGF receptor 1 (svegfr1)/ soluble Fms Related Receptor Tyrosine Kinase 1 (sflt1))* pathway components. We show increased phospho-ERK staining in migrating angioblasts, consistent with enhanced Vegf activity. Reducing Vegfr2 kinase activity in *sema3fb* mutants rescues angiogenic sprouting. Our data suggest that Sema3fb plays a critical role in promoting endothelial sprouting through modulating the VEGF signaling pathway, acting as an autocrine cue that modulates intrinsic growth factor signaling.

## Author summary

To supply tissues with essential oxygen and nutrients, blood vessel development is carefully orchestrated by positive 'go' and negative 'stop' growth signals as well as directional cues to shape patterning. Semaphorin proteins are named after the 'Semaphore' railway signaling system that directed trains along the appropriate tracks. Our work highlights the role of the Semaphorin 3fb protein in providing a pro-growth signal to developing vessels. Semaphorin 3fb is both produced by, and acts on the precursors of blood vessels as they migrate, a process known as autocrine control. We find that losing Semaphorin 3fb leads to stalled blood vessel growth, indicating it normally acts as a positive signal. It acts via

iac01.ucalgary.ca/FGSA/Public/SpecificAward.
aspx?AwardID=5956), and RH was supported by
T. Chen Fong Graduate studentship (https://iac01.
ucalgary.ca/FGSA/Public/SpecificAward.aspx?
AwardID=60410) and Alberta Innovates-Health
Solutions studentship (https://albertainnovates.ca/
programs/graduate-studentships-in-health-
innovation/). Operating funding was provided by a
CIHR Project grant funding (https://cihr-irsc.gc.ca;
PJT-168938) to SJC, and Brightfocus Foundation
(https://support.brightfocus.org), and CIHR Project
grant funding (https://cihr-irsc.gc.ca;PJT-153062)
to SM. The funders had no role in study design,
data collection and analysis, decision to publish, or
preparation of the manuscript.

**Competing interests:** The authors have declared
that no competing interests exist.

modulating the VEGF growth factor signaling pathway that in turn, controls the migration process. We propose that Semaphorin3b fine-tunes vessel growth, thus ensuring a properly patterned network develops.

## Introduction

Sprouting angiogenesis is a process by which new vessels branch and grow from existing vessels, establishing the perfusion of tissues and organs. How sprouting angiogenesis is coordinated at the intrinsic and extrinsic levels is one of the most important questions in vascular biology. Vessel growth is highly dependent on Vascular Endothelial Growth Factor (VEGF) signaling, which functions as a master regulator to promote angiogenesis, and is often dysregulated in disease [1–4]. VEGF gradients within tissues are responsible for the initial triggering and guidance of the sprouting process signaling through the endothelial-expressed tyrosine kinase VEGF receptor 2 (VEGFR2) [5–7], activating downstream signaling including that of the MAPK-ERK pathway. The cellular response to VEGF must be carefully regulated to ensure the stereotypical patterning of vessels.

During sprout formation, angioblasts adopt two distinct cellular states–termed tip and stalk- that respond to internal and external stimuli to promote and guide vessel growth [5,8,9]. Tip cells utilize filopodia to scan the environment for attractive and repulsive cues that dynamically control proliferation and migration [10,11]. In contrast, trailing stalk cells have limited filopodia, are quiescent, and contribute to forming the vascular lumen and phalanx [5,12]. Tip and stalk identity are determined competitively through a Delta-like 4 (Dll4)-Notch lateral inhibition pathway and VEGF signaling feedback loop. Dll4 expression is induced downstream of VEGF signaling in tip cells [13]. Dll4 positive cells activate the Notch receptor in stalk cells [14–16] to down-regulate VEGF receptor expression and limit the response of the stalk cell to environmental VEGF [17–19].

The vertebrate-specific secreted Class 3 Semaphorins (Sema3s) including Sema3a, Sema3c, and Sema3f typically act as repulsive guidance cues to limit vessel growth. Sema3 control of angiogenesis is evolutionarily conserved across multiple species [20–23]. The exact role of these instructive cues can differ depending on the species and tissue expression of ligands and/ or receptors. PlexinD1 is an endothelial-specific receptor conserved in zebrafish and mouse and integrates signals from multiple Sema3 ligands. During embryonic vessel growth, zebrafish Sema3a is expressed in a caudal-to-rostral gradient across each somite. Paracrine signaling from Sema3a to its receptor PlexinD1 in the endothelium acts to spatially restrict intersegmental vessels [22]. In mouse Sema3E guides intersegmental vessel growth via PlexinD1 [20,24]. In contrast, fish Sema3e shows endothelial expression and acts as an autocrine pro-angiogenic factor to antagonize PlexinD1 during intersegmental vessel growth [25]. PlexinD1 also limits responses to VEGF by increasing the expression of a soluble decoy Vegf receptor (sVegfr2/ sFlt1) [26]. sFlt1 antagonizes Vegfr2 signaling by sequestering Vegf ligand thus providing a link between Sema-Plexin and Vegf signaling.

Sema3F is highly expressed by cultured human and mouse endothelial cells [27–30] and in endothelial cells in scRNAseq databases [31,32]. SEMA3F typically signals through Neuropilin receptors (NRP) to modulate migration in cell culture models [30,33]. Exogenous human SEMA3F inhibits tumor progression and vascularization [21]. However, SEMA3F has a versatile role in regulating vessel growth depending on context; it functions either as a pro-angiogenic or anti-angiogenic cue. For instance, exogenously applied Sema3F limits aberrant growth of mouse retinal vessels [34], and we recently showed similar anti-angiogenic functions

for *sema3fa* in the zebrafish retina [35]. At the same time, Sema3F is pro-angiogenic during placental development in mice [29,36]. There is still a limited understanding of how an endothelial cell can receive the same signal, yet coordinate different downstream molecular pathways driving angiogenic growth. Context-dependent and tissue specific expression can further influence cell behaviors, for example in the mouse brain Sema3E/PlexinD1 signaling switches from attractive to repulsive depending on NRP co-receptor expression [37].

Here we investigate the role of zebrafish Sema3f in regulating angiogenic growth. We find that *sema3fb* is expressed within endothelial cells of the dorsal aorta and is necessary for promoting sprout migration in the trunk. We show that *sema3fb* regulates the expression of the VEGF receptor *vegfr2* to promote angioblast migration, as well as regulating the expression of genes mediating feedback on Vegf signaling such as *sflt1* and *dll4*. Together these data reveal a new autocrine role for a secreted ligand from endothelial cells in modulating VEGF pathway activity in angiogenic sprouting.

## Results

### *sema3fb* is expressed by developing blood vessels

To investigate the role of Sema3F in vascular angiogenesis, we use zebrafish. The zebrafish genome contains duplicated orthologs of human *SEMA3F* - *sema3fa* and *sema3fb* [38]. The stereotypic patterning of major trunk vessels in the zebrafish begins at around 20 hours post fertilization (hpf) when angioblasts (endothelial precursors) migrate collectively from the dorsal aorta (DA) and sprout laterally in between each pair of somites to form the intersegmental arteries (ISAs). The ISAs then migrate dorsally and connect to form the dorsal longitudinal vessel (DLAV) between 30–32 hpf [39]. Using whole mount in situ hybridization (ISH), we analyzed the expression of *sema3fa* and *sema3fb* and find non-overlapping patterns of expression in the trunk of developing embryos. *sema3fb* is expressed in the dorsal aorta at 26 hpf and in ISAs by 28–30 hpf (Figs 1A and S1), similar to the endothelial expression pattern of *sema3e* [25]. In contrast, its homolog *sema3fa* is absent from the trunk endothelium and is expressed laterally in somites (S1 Fig). These expression patterns are consistent with published zebrafish single-cell sequencing data showing expression of *sema3fb* (but not *sema3fa*) in *pecam*, *tie1*, and *flt1*-expressing endothelial cells of developing zebrafish embryos [40]. Endothelial expression of Sema3f in murine and human and of *sema3fb* in zebrafish vasculature is supportive of the conserved role for Sema3f in endothelial cells.

### Loss of *sema3fb* results in angiogenic deficits

To investigate the endogenous role of Sema3fb in regulating vessel growth in an intact animal, we used the *sema3fb*$^{ca305}$ CRISPR mutant with a 19 base pair deletion in exon1 that is predicted to produce a premature truncated protein (32 amino acids in length), deleting most of the Sema domain which is necessary for intracellular signaling [41,42]. To visualize real-time vascular development, we crossed *sema3fb*$^{ca305}$ loss of function mutants to *Tg(kdrl:mCherry)*$^{ci5}$ transgenic fish that fluorescently mark endothelial cells, and generated wild type, heterozygote, and mutant siblings. We analyzed angiogenic sprouting at 30hpf when ISA sprouts first connect to form the DLAV (Fig 1B) and observed angiogenic deficits in both *sema3fb*$^{ca305/+}$ heterozygotes (Fig 1D) and *sema3fb*$^{ca305}$ homozygous mutants (Fig 1E) as compared to wild type siblings (Fig 1C). Specifically, we note a significant reduction in the average length of the ISA sprouts at 30 hpf from 106 μm in wild type to 91 μm and 92 μm in heterozygote and homozygote mutant embryos, respectively (Fig 1F). Second, the percentage of ISAs connected at the DLAV is reduced from 80% in wild type to 50% in heterozygote and 40% in homozygote mutants (Fig 1G). Lastly, ventral sprout diameter is increased from 7 μm in wild type to

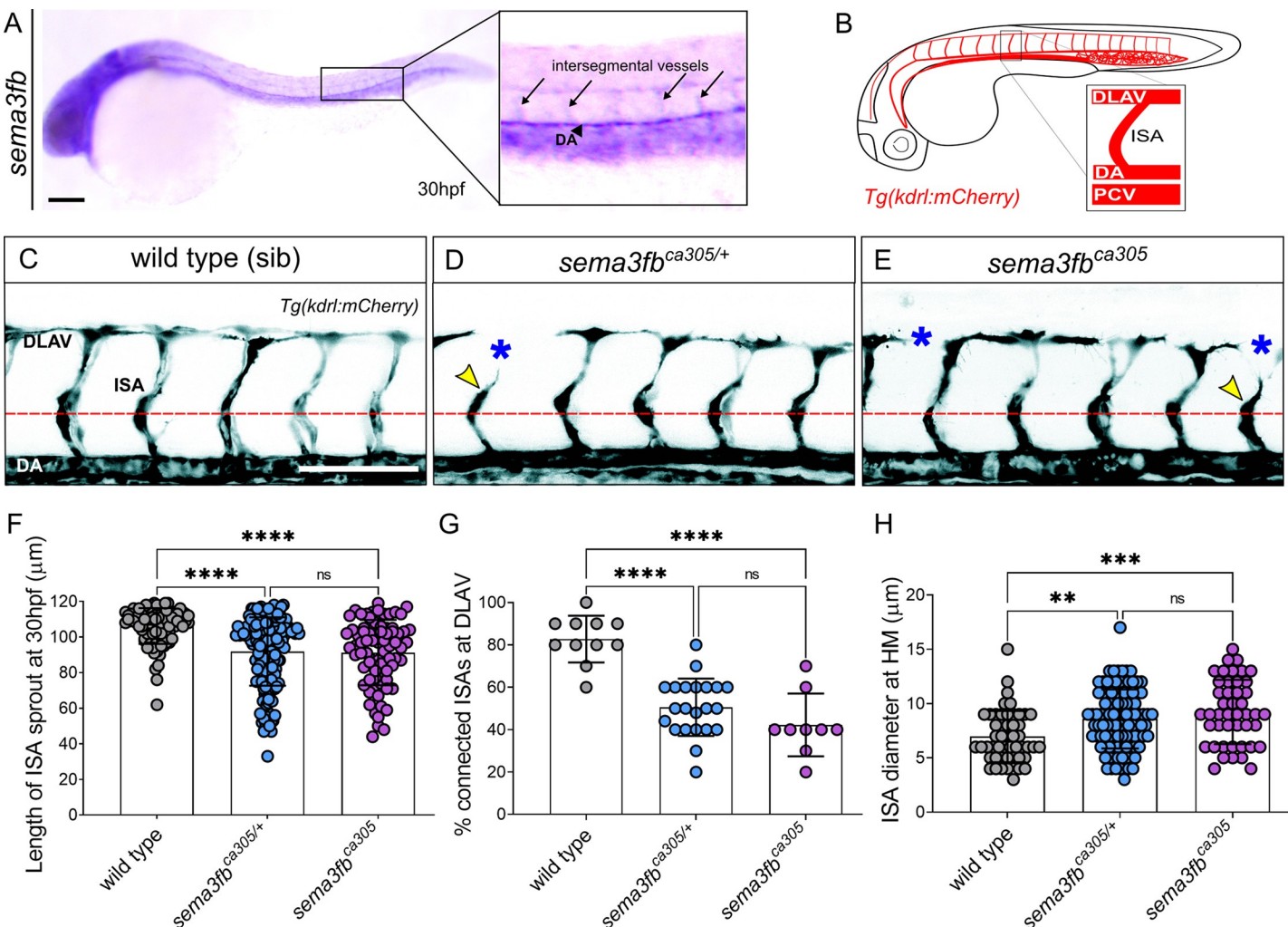

**Fig 1. Endothelial expressed sema3fb promotes endothelial cell sprouting.** A) Lateral view of sema3fb expression at 30hpf by ISH. Inset shows expression in the dorsal aorta (DA) and intersegmental arteries (ISAs). B) Schematic representation of the zebrafish vasculature at 30 hpf. Inset: The ISAs sprout from the DA and connect to form the Dorsal Longitudinal Anastomotic Vessel (DLAV) by 30 hpf. C-E) Lateral confocal images of the trunk vasculature (black) of 30 hpf (C) wild type sibling (sib), (D) heterozygous (het) sema3fb^ca305/+ and (E) homozygous (hom) sema3fb^ca305 mutants. Gaps in the DLAV (blue asterisks) and truncated ISA sprouts (yellow arrowhead) are noted. Abbreviations: DA (Dorsal Aorta), and PCV (Posterior Cardinal Vein). Anterior, left; Dorsal, up. Scale bar, 100 μm. F) ISA Sprout length at 30 hpf in wild type (WT) sibs (mean length of 106±10 μm), het sema3fb^ca305/+ (92±19μm), and hom sema3fb^ca305 (91±18μm), ****p<0.0001. G) Percentage of ISA sprouts connecting to form DLAV at 30 hpf. WT sib (mean 83% connected), het sema3fb^ca305/+ (50% connected) and hom sema3fb^ca305 (42% connected), ****p<0.0001. H) Quantification of the cross-sectional diameter of ISA sprout at the level of the horizontal myoseptum (HM; dashed red line). WT sib (mean diameter of 6.9±2 μm), het sema3fb^ca305/+ (8.5±2.7 μm), and hom sema3fb^ca305 (9.2±2.9 μm)., **p<0.0016 and ***p = 0.0002. F-G) N = 3; WT sib = 11 embryos (n = 86 ISAs), het sema3fb^ca305/+ = 22 embryos (n = 163 ISAs), and hom sema3fb^ca305 = 9 embryos (n = 75 ISAs), 2-Way ANOVA Tukey's multiple comparisons test. Error bars = ±SD.

8.5 μm in heterozygotes and 9 μm in homozygotes (Fig 1H). These data suggest Sema3fb normally promotes angiogenic migration, resulting in stunted wider sprouts when lost.

We observed that losing a single mutated allele of *sema3fb* results in highly penetrant vascular defects in the developing trunk, with a similar magnitude to homozygous loss of the gene. This could result if the loss of function of *sema3fb* is haploinsufficient. To induce a loss of function by an independent method, we injected 1-cell stage embryos with 1ng of a validated morpholino antisense oligonucleotide (morpholino, MO) against *sema3fb* [43]. *sema3fb* morphants have an identical phenotype to *sema3fb*^ca305 mutants at 30 hpf, with no additional defects in morpholino-injected *sema3fb*^ca305 mutants suggesting that the sema3fb mutant is a

loss of function allele (S1 Fig). As a note, haploinsufficiency is also observed for members of the Vegf pathway including the Vegf and Dll4 genes, both critical for angiogenesis.

As the *sema3fb* paralog *sema3fa* is not expressed in trunk vasculature (S1 Fig), we did not expect the loss of *sema3fa* to affect ISA growth. Indeed, *sema3fa^ca304^* mutants display normal sprouting and connections (S1 Fig). There also does not appear to be genetic compensation by *sema3fa*, as *sema3fa^ca304^* mutants injected with *sema3fb* morpholino show no additional defects over *sema3fb^ca305^* mutants or morphants (S1 Fig). These results suggest that only one of the two zebrafish Sema3f orthologs, *sema3fb*, regulates angiogenic sprouting in the trunk at this developmental stage.

Despite Semaphorin involvement in axon and angioblast guidance in other systems, we find no significant changes in sprout direction between *sema3fb* heterozygotes or homozygous mutants as observed by the position of sprouts with reference to laminin staining at the myo-tendinous junctions that separate somites (S2 Fig). Like other angiogenic mutants with stalled migration [44], *sema3fb* mutant vessels eventually recover, likely through compensation by unknown genes and pathways, with no noticeable differences in vessel patterning at 48 hpf (S3 Fig).

We did not expect blood flow to contribute to *sema3fb* mutant ISA phenotypes as the migration of cells forming primary ISAs between 20–30 hpf occurs independently of blood flow [45,46]. However, as *sema3fb^ca305^* mutants have a heart function defect [41] we next tested whether there were effects of blood flow on the phenotype. We injected 1ng of a commonly used translation blocking morpholino against cardiac *troponin 2a* (*tnnt2aMO;* [47]) in wild type and *sema3fb* mutants at the one-cell stage, to limit heart contractility and blood flow. In *tnnt2* MO-injected wildtypes, ISA growth is unaffected by the loss of blood flow as previously reported. Similarly, there is no significant difference in vessel growth or numbers of connections in the *tnnt2a* MO-injected *sema3fb* mutants as compared to their uninjected counterparts (S2 Fig). These data demonstrate the angiogenic defects in *sema3fb* mutants at 30 hpf are not a result of modified blood flow. We note that uninjected *sema3fb^ca305^* fish show a reduction in the DA diameter (S2 Fig), an effect also seen in *tnnt2a* morphants. This suggests that axial vessel diameter is sensitive to cardiac output at early time points while sprouting angiogenesis is not. Thus, our data to this point suggest that Sema3fb normally promotes ISA sprouting from the dorsal aorta, with no apparent role in spatial guidance.

## Loss of *sema3fb* disrupts ISA migration and endothelial nuclei morphology

The stunted morphology of s*ema3fb^ca30^* ISA vessels at 30 hpf suggests there might be delays in EC sprout initiation and/or migration. To examine whether these are impacted by the loss of Sema3fb, we crossed *sema3fb^ca305^ Tg(kdrl:mCherry)* fish to the *Tg(fli1a:nEGFP)* line, which labels endothelial nuclei (Fig 2A). Using time-lapse confocal imaging we tracked ISA vessel elongation between wild type and *sema3fb^ca305^* embryos from 25–29 hpf (Fig 2A and 2B). We find no significant difference in sprout initiation from 25.0–25.5 hpf in either wild type or *sema3fb^ca305^* mutant ISAs, both having an average length of 54 μm. However, as sprouts begin to elongate, the average ISA migration distance for mutants starts to fall behind wild type distances by 26. 5 hpf and vessels are shorter at all subsequent time points analyzed, with an average 5–9 μm difference in length between wild type and mutant (Fig 2B and S1 Table). ISA angioblast migration speed from 25–26 hpf averages 0.21 μm/min (S4 Fig) and slows to 0.15 μm/min between 26–27 hpf (Fig 2C) before increasing to 0.19 μm/min by 27–28 hpf (Fig 2D) and continues at 0.16 μm/min between 28–29 hpf (Fig 2E). In contrast, the average rate of migration for *sem3fb^ca305^* vessels is significantly reduced to an average of 0.12–0.13 μm/min from 25–28 hpf when compared to wild type (Figs 2C, 2D and S4). We also measured the lead

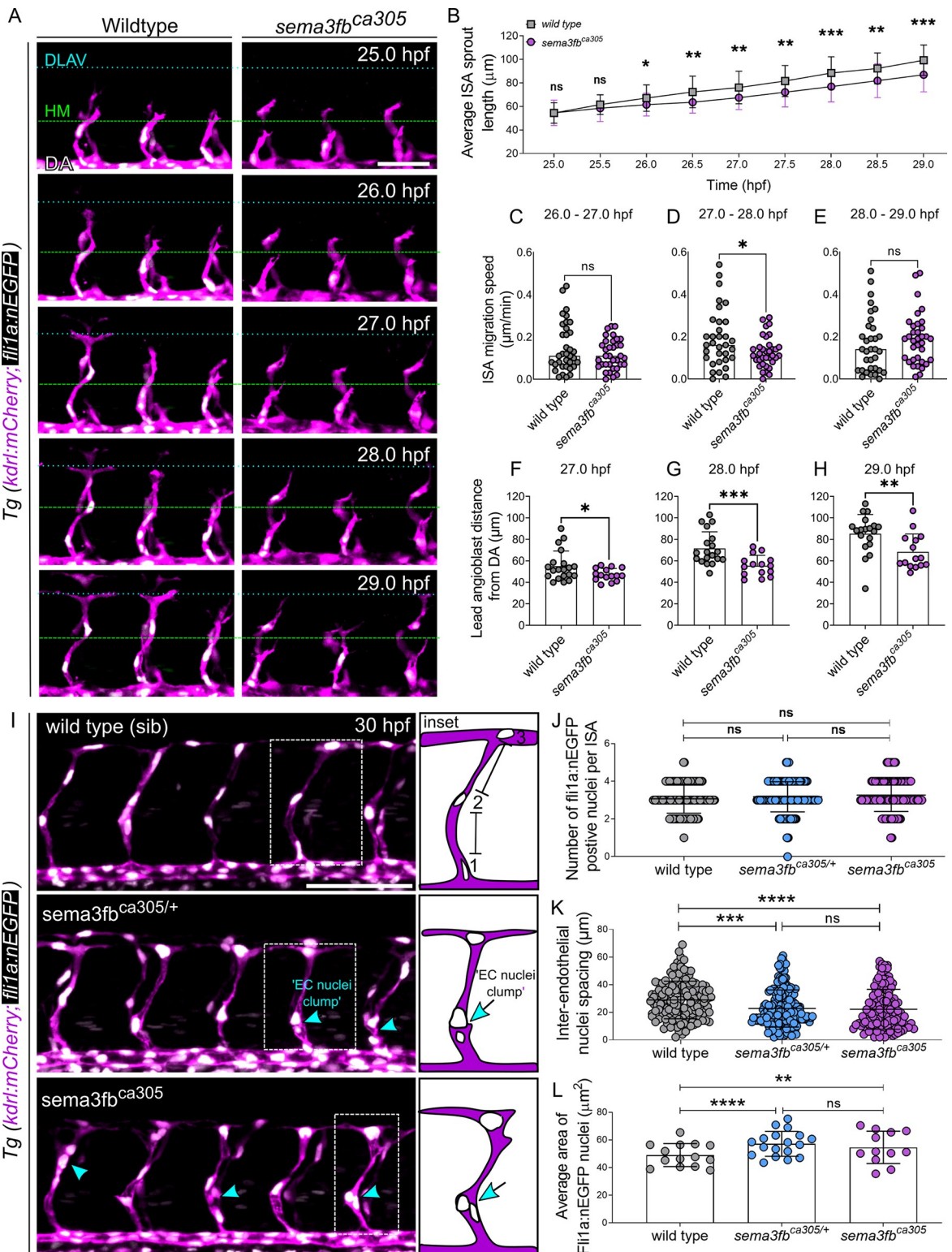

**Fig 2. Loss of *sema3fb* disrupts ISA migration.** A) Lateral confocal time-lapse images from 25–29 hpf double transgenic Tg(kdrl:mCherry;fli1a:nEGFP) endothelial cells (magenta) and nuclei (white). The location of the horizontal myoseptum (green dashed line) and DLAV (blue dashed line) are noted to highlight ISA growth over time. Scale bar, 50 μm. B) Average ISA Sprout Length at 30-minute intervals from 25–29 hpf: WT vs *sema3fb*^ca305 at 25.0 hpf, p = 0.474; at 25.5 hpf p = 0.262; at 26.0 hpf p = 0.081; at 26.5 hpf *p = 0.023; at 27.0 hpf *p = 0.020; at 27.5 hpf **p = 0.030; at 28.0 hpf **p = 0.008; at 28.5 hpf **p = 0.007; at 29.0 hpf *p = 0.024. C-E) Quantification of

ISA migration speeds (μm/min). C) At 26–27 hpf WT = 0.15 μm/min and *sema3fb*ca305 = 0.12 μm/min, p = 0.157. D) At 27–28 hpf WT = 0.19 μm/min and *sema3fb*ca305 = 0.13 μm/min, *p = 0.020. E) At 28–29 hpf: WT = 0.16 μm/min and *sema3fb*ca305 = 0.19 μm/min, p = 0.461. B-E) N = 2; WT = 7 embryos (n = 33 ISAs) and *sema3fb*ca305 = 7 embryos (n = 35 ISAs), Unpaired t-test with Welch's correction. F-H) Lead angioblast distance from DA at 1hr intervals. F) At 27 hpf mean distance from DA: WT = 55.12±14.06 μm and *sema3fb*ca305 = 47.18±5.75 μm, *p = 0.030. G) At 28 hpf mean distance from DA: WT = 71.57±15.47 μm and *sema3fb*ca305 = 55.47 ±9.65 μm, ***p = 0.0008. H) At 29 hpf mean distance from DA: WT = 85.19±18.03 μm and *sema3fb*ca305 = 68.36±16.64 μm, **p = 0.008. F-H) N = 1: WT = 4 embryos (20 ISAs) and *sema3fb*ca305 = 3 embryos (15 ISAs), Unpaired t-test with Welch's correction. I) Lateral confocal images of 30hpf double transgenic Tg(kdrl:mCherry;fli1a:nEGFP) endothelial cells (ECs, magenta) and nuclei (white). EC nuclei clumps (blue arrows/arrowheads) are noted. Scale bar, 100 μm. Inset: Schematics show method for measuring distance between EC nuclei and highlight EC nuclei clumps in ISAs. J) Number of EC nuclei (angioblasts) per ISAs at 30 hpf. WT (mean of 3 nuclei/ISA), heterozygous (het) *sema3fb*ca305/+ (3 nuclei/ISA), and homozygous (hom) *sema3fb*ca305 (3 nuclei/ISA). K) Quantification of inter-endothelial nuclei spacing per ISA at 30 hpf. WT (mean 28±13 μm), het *sema3fb*ca305/+ (23±13μm), and hom *sema3fb*ca305 (22±14 μm), ***p = 0.0002 and ****p<0.0001. L) Quantification of Average Area EC fli1a:nEGFP positive nuclei (angioblasts) per ISAs at 30 hpf. WT (mean 49±18 μm²), het *sema3fb*ca305/+ (mean 60±24 μm²), and hom *sema3fb*ca305 (mean 56±23 μm²),**p = 0.0069 and ****p<0.0001. I-L) N = 3; WT = 14 embryos (n = 138 ISAs), het *sema3fb*ca305/+ = 19 embryos (n = 190 ISAs), and hom*sema3fb*ca305 = 11 embryos (n = 110 ISAs),. 2-Way ANOVA Tukey's multiple comparisons test Error bars = ±SD.

angioblast (tip cell) distance from the DA at hourly intervals as a measure of EC motility and similarly find no difference in the distance traveled between wild type and mutant vessels at 25–26 hpf (S4 Fig). There is a significant difference by 27hpf, with an average distance traveled of 55 μm for wild type at 27 hpf (Fig 2F) to 70 μm at 28 hpf (Fig 2G), while mutants only reach 47 μm and 55 μm respectively. By 29 hpf mutants migrate an average of 69 μm from the DA whereas wildtype angioblasts migrate significantly further and reach 85 μm (Fig 2H). Together these data suggest that sprout initiation is normal, but angioblast movement is significantly delayed with loss of *sema3fb*.Alongside the stunted morphology, we observed that loss of sema3fb resulted in wider vessels (Fig 1H). Previous studies in zebrafish have linked increased ISA width to changes in EC proliferation [48–50]. To determine whether changes in cell number could account for the change in vessel diameter, we assayed ISA sprout formation in the *Tg(kdrl:mCherry)*; *(fli1a:nEGFP)* lines (Fig 2I). ISA growth follows a stereotypic series of angioblast movements initiated by the migration of an angioblast tip cell from the dorsal aorta followed by a second trailing stalk cell. Once the leading angioblast reaches the horizontal myoseptum, it typically divides and a single daughter cell migrates dorsally to form the DLAV [25,51,52]. This process results in an average of 3–4 endothelial cell nuclei per ISA sprout (Fig 2B). *sema3fb*ca305 mutants and heterozygotes have a similar number of endothelial cells per vessel as wild type animals (Fig 2J). *sema3fb* morphants also have the same number of endothelial nuclei per vessel as wild type (S4 Fig), suggesting that loss of *sema3fb* does not affect endothelial cell number.

Analysis of the distance between endothelial nuclei, however, revealed a significant decrease in the spacing between endothelial cell nuclei in *sema3fb* mutants (Fig 2I and 2K). Reduced spacing gives the appearance of nuclear clumps. Clump location below the horizontal myoseptum correlates with an increased width of ISAs (Fig 1H). We also find that endothelial nuclei are significantly larger in mutants than in wild type controls, increased from an average of 49 μm² in wild type to 60 μm² heterozygous and homozygous mutants (Fig 2L). The increase in nuclear size is also observed in *sema3fb* morphants (S4 Fig). Taken together, this data suggests that *sema3fb* mutant sprouts have the correct number of cells, can migrate along the correct path, but do not migrate as far as wild type sprouts. By 30hpf, stalled growth disrupts ISA elongation resulting in aberrant cell and nuclear morphology.

## *sema3fb* mutants display persistent filopodia

As Sema3F signaling controls actin dynamics, we reasoned that angioblast migration defects might also be accompanied by a change in filopodia activity. As ISAs begin to connect by 28

hpf, through a process termed anastomosis, actin-rich filopodia projections resolve to allow stable vessel connections forming a patent DLAV. In *sema3fb*<sup>ca305</sup> mutants, a small proportion of ISAs can reach and connect to form the DLAV by 30 hpf (Fig 1G). To assay whether loss of *sema3fb* can impact the vessel connection process, we injected wild type and *sema3fb*<sup>ca305</sup> mutants with an endothelial-specific F-actin reporter, Tol2(*fli1*<sup>ep</sup>*Lifeact-EGFP*), in which filamentous actin (F-actin), a major component of the cytoskeleton, is labeled with GFP to visualize filopodia on migrating endothelial cells (Figs 3A and S5). We selected GFP positive ISA sprouts that have reached the level of the DLAV, and assayed filopodia above the horizontal myoseptum but below the DLAV (S5 Fig) to capture filopodia projections along the upper segment of each ISA. These filopodia should rapidly disappear once the ISV is connected from around 28 hpf. Time-lapse confocal imaging of mosaically labeled sprouts from 28–30 hpf in stage-matched embryos revealed that wild type sprouts have filopodia when they first reach the DLAV but filopodial numbers gradually reduce over time as the cells lumenize and become quiescent once connected to neighboring vessels (Figs 3B and S5). At 28hpf, wild type and *sema3fb* mutants have the same number of filopodia per ISA (average of 9 filopodia). By 29hpf, wild type sprouts have an average of 5 filopodia, while sema3fb mutants have 7. By 30hpf, there is an average of 4 filopodia remaining in wild type, while *sema3fb* mutants maintain comparable numbers to the earlier time points (Fig 3B and 3C). These data suggest that in the few sprouts of *sema3fb* mutants that reach the DLAV, cells fail to restrict filopodia formation at the appropriate time.

## VEGF signaling is altered in *sema3fb* mutants

To investigate the molecular mechanisms controlling migration downstream of *sema3fb* we assayed gene expression in FACS-sorted *Tg(kdrl:mCherry)* wildtype and *sema3fb* mutant endothelial cells by Taqman RT-qPCR. We analyzed the expression of a key set of endothelial cell markers that regulate angiogenic sprouting (Figs 4A and S6). Analysis of FACs-sorted endothelial cells reveals a 4.3-fold increase in *vegfr2/kdrl* and a 2.3-fold increase in *dll4* in *sema3fb* mutant endothelial cells as compared to controls. Comparison of the fluorescence intensity of transgenic *Tg(kdrl:mCherry)* also revealed a significant increase in mCherry fluorescence in live sema3fb<sup>ca305</sup> mutants (S6 Fig). We find no significant difference in *notch2* or *jagged1a* or the *vegfr2* ligand *vegfab* expression (Fig 4A). Surprisingly, we also find a significant near 3-fold increase in the expression of the decoy receptor *soluble vegfr1/soluble flt1 (sflt1)* in mutants as compared to controls (Fig 4B). As an independent experiment, we confirmed upregulated *vegfr2* and *sflt1* expression in mutants using quantitative fluorescent ISH (FISH) (Fig 4B and 4C). We next assayed the expression of Phospho-ERK (pERK), a downstream effector of VEGF signaling in vascular sprouting. Although there is no difference in the number of ISAs with a positive pERK signal between wildtype and mutant (Fig 4D and 4E), we find that there is a significant increase in fluorescent intensity of pERK positive EC nuclei in mutant ISV angioblasts (Fig 4D and 4F), which is supportive of increased Vegfr2 pathway activity in ECs (Fig 4D and 4E). These data suggest the loss of Sema3fb results in upregulation of both VEGF-pathway promoting (*vegfr2/dll4*) and VEGF-inhibiting (*sflt1*) genes in endothelial cells, which could account for disrupted angioblast migration.

To test whether the angiogenic defects in *sema3fb* mutants are the result of increased Vegfr2 activity, we treated wild type and *sema3fb* mutant embryos with 0.5 μM SU5416, a selective Vegfr2 inhibitor, from 20–30 hpf (Fig 4G). Inhibitor-treated wild type embryos have stunted sprouts as reported (104 μm in untreated wildtype vs. 50 μm in treated wild type embryos), but *sema3fb* mutants are unaffected with no significant changes in sprout length (average length 82 μm in both treated and untreated mutants; S6 Fig). This suggests that

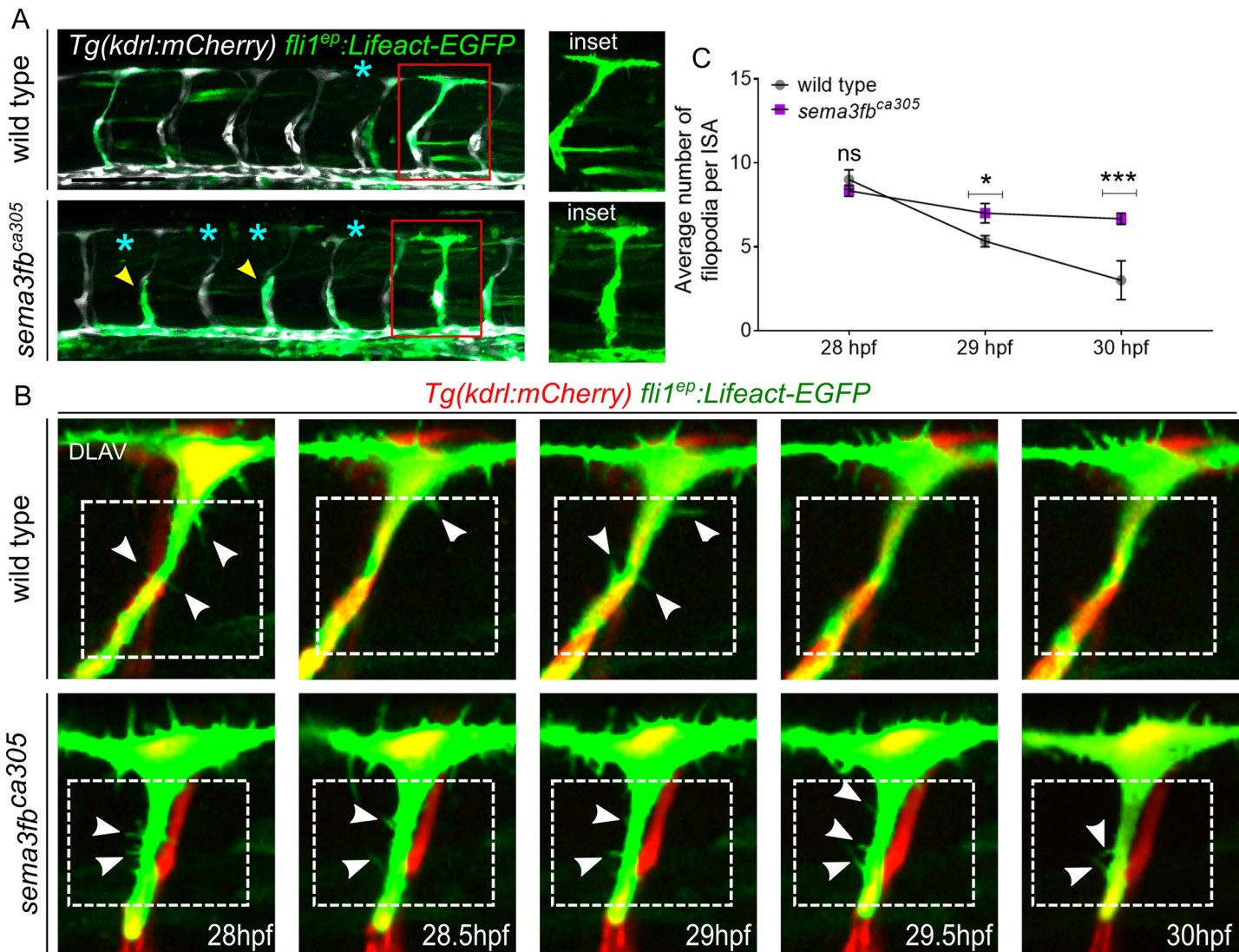

**Fig 3. sema3fb mutants display aberrant and persistent filopodia in the dorsal ISA.** A) Lateral images of the trunk vasculature with mosaic endothelial expression of the transgene fli1ep: Lifeact-EGFP highlighting actin (green) and endothelial cytoplasm using Tg(kdrl:mCherry; white) in ISAs at 30 hpf. DLAV gaps (blue asterisks) and truncated ISA sprouts (yellow arrowheads) are marked. Inset shows an enlarged view of single ISAs with Lifeact-EGFP expression that have reached the level of DLAV at 30hpf. Scale bar, 100 μm. B) Representative still images from time-lapse imaging from 28–30 hpf. Enlarged still images of stage-matched embryos with mosaic Lifeact-EGFP (green) in endothelial cells spanning the ISA and reaching the level of the DLAV by 28 hpf in both wild type and sema3fb[ca305] embryo. Endothelial cytoplasm is shown in red Tg(kdrl:mCherry). White arrowheads indicate filopodia present in connecting ISA sprouts within the boxed regions below the DLAV. C) Quantification of number Lifeact-EGFP positive filopodia on ISAs from 28–30 hpf from embryos of the indicated genotypes. N = 3: WT (14 EGFP positive ISAs/ 30 ISAs total, 6 embryos, mean of 4 filopodia/ISA) and homozygous sema3fb[ca305] (18 EGFP positive ISAs/35s ISAs total, 7 embryos, mean of 7 filopodia/ISA). Unpaired t-test with Welch's correction,*p = 0.03 and ***p = 0.0002. Error bars = ±SD.

increased *vegfr2* expression in *sema3fb* mutants renders them less sensitive to pharmacological Vegfr2 inhibition as compared to wild type embryos. However, we reasoned that this inhibitor dose severely disrupts VEGF signaling and vessel growth and may mask more subtle changes in sprout dynamics. Therefore, we lowered the SU4516 concentration to a sub-optimal dose that others have used (low-dose: 0.2μM SU5416) to induce minimal angiogenic deficits (Fig 4H) [53]. As expected, in wild type embryos low-dose treatment results in a mild reduction in sprout length from 104 μm to 92 μm and a 60% decrease in the number of ISA connections to the DLAV (Fig 4I and 4J). However, in *sema3fb* mutants, sub-optimal Vegf inhibition

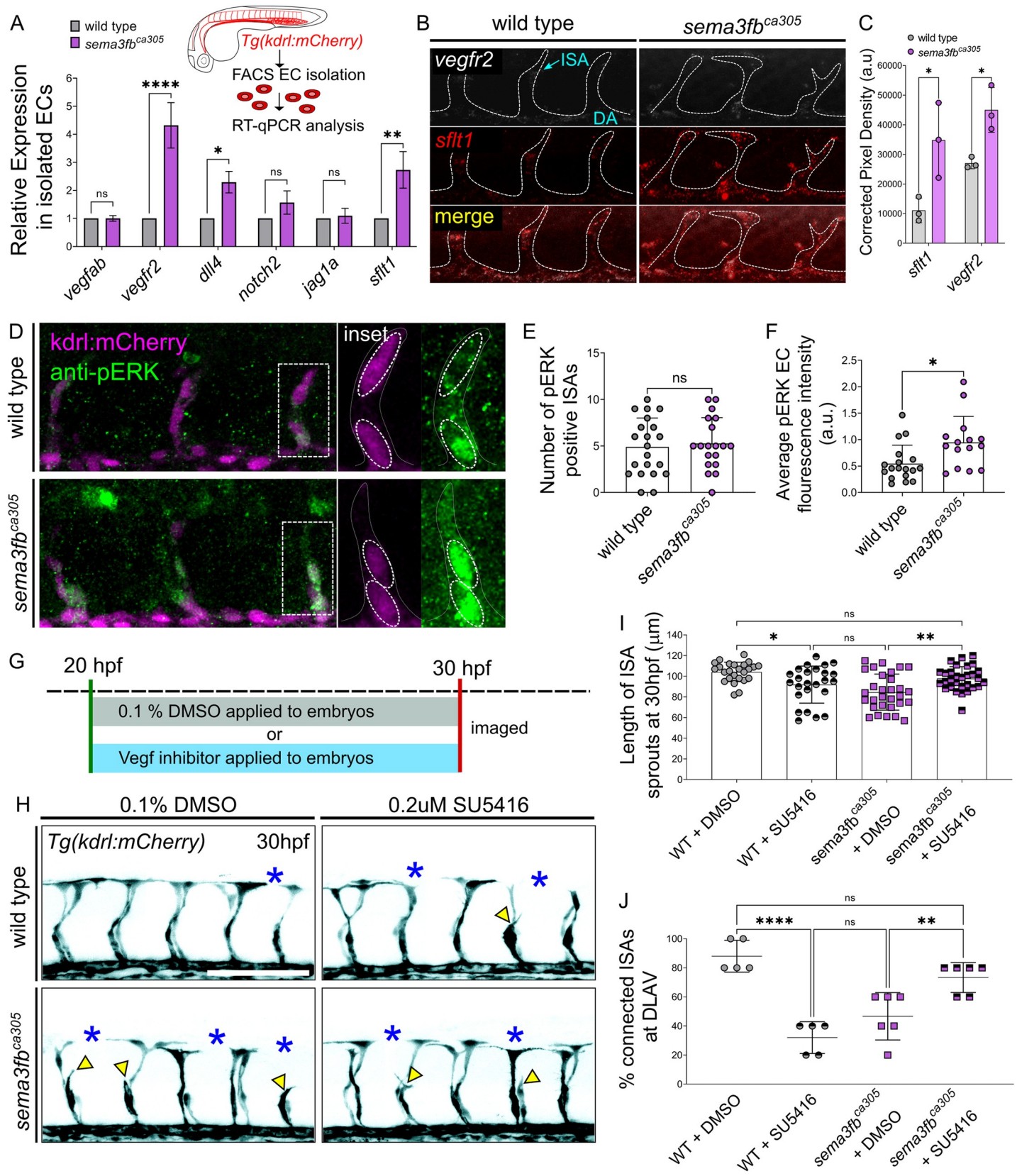

**Fig 4. *sema3fb* mutants have increased VEGF receptor expression and activity.** A) RT-qPCR analysis of key endothelial markers in wild type and *sema3fb*ca305 FACS isolated Tg(kdrl:mCherry) positive endothelial cells at 26hpf (inset). N = 2, 2-Way ANOVA Tukey's multiple comparisons test, *p = 0.0184, **p = 0.0021, and ****p<0.0001 (Refer to S3 Table for fold-change details). B) Fluorescent HCR in situ of 30 hpf whole-mount wild type and embryos *sema3fb*ca305 embryos. Representative images show punctate overlapping expression of *vegfr2 (white) and sflt1 (red)* mRNA transcripts within the DA and ISAs (dashed white outline). C) Quantification of HCR in situ pixel density in ISAs and DA, wild type (WT, n = 3 embryos, 15 ISAs) and *sema3fb*ca305 (n = 3 embryos, 15 ISAs), Unpaired Student's t-test with Welch's correction WT vs. *sema3fb*ca305: *vegfr2* *p = 0.047 and *sflt1* *p = 0.036. D) Whole-mount Immunostaining for phosphoERK (pERK) in WT and *sema3fb*ca305 embryos fixed at 30 hpf. Representative images show *Tg(kdrl:mCherry)* positive ISAs (purple) and pERK positive ECs (green). Inset: pERK positive ISAs are traced using *kdrl:mCherry* expression (dashed white line) and dashed oval outlines highlight individual ECs with pERK staining within each ISA. E) Number of pERK positive ISAs at 30hpf. F) Quantification of average pERK fluorescence intensity in embryos at 30 hpf. D-E) N = 3, WT (n = 21 embryos, mean of 5 pERK positive ISAs), and homzygous *sema3fb*ca305 (n = 19 embryos, mean of 5 pERK positive ISAs). 2-Way ANOVA Tukey's multiple comparisons test, *p = 0.012. G) Schematic of Vegfr2 inhibition time course, embryos are treated at 20 hpf with either 0.1%DMSO or Vegfr2 inhibitors and removed from treatment for live imaging at 30hpf. H) Representative confocal images of trunk vasculature (black) of 30 hpf embryos treated with DMSO control or 0.2 μM SU5416. DLAV gaps (blue asterisks) and truncated ISA (yellow arrowheads) are marked. Scale bar, 100 μm. I) Length of ISA sprouts in treated embryos at 30 hpf: WT + DMSO (n = 25 ISAs, mean of 104±9 μm), WT + 0.2 μM SU5416 (n = 25 ISAs, mean of 92±17 μm), *sema3fb*ca305 + DMSO (n = 30 ISAs, mean of 85±17 μm), and *sema3fb*ca305 + 0.2μm SU5416 (n = 30 ISAs, mean of 98±11 μm) **p = 0.0039 and ****p<0.0001. J) Percentage of ISA sprouts connected to DLAV in treated embryos at 30 hpf. WT + DMSO (n = 25 ISA-DLAV, 5 embryos, mean 88±11%), WT + 0.2μM SU5416 (n = 25 ISA-DLAV, mean 32±11%), *sema3fb*ca305 + DMSO (n = 30 ISA-DLAV, mean 46±16%), and *sema3fb*ca305 +0.2 μm SU5416 (n = 30 ISA-DLAV, mean 73±10%), **p = 0.0084, ***p = 0.0002, and ****p<0.0001. I-J N = 2; WT+DMSO = 5 embryos, WT + 0.2 μM SU5416 = 5 embryos, *sema3fb*ca305 + DMSO = 6 embryos, *sema3fb*ca305 + 0.2 μm SU5416 = 6 embryos,. 2-Way ANOVA Tukey's multiple comparisons test. Error Bars = ±SD.

significantly rescues *sema3fb* mutant defects. The length of ISAs in treated *sema3fb* mutants increases to 98 μm from an average of 85 μm in untreated mutants, restoring them to an ISA length indistinguishable from wildtype (Fig 4I). Connections at the DLAV are also significantly increased from 46% in untreated mutants to 73% in treated embryos, which is comparable to untreated wild types and significantly increased beyond treated wildtypes. (Fig 4J). We also applied a second VEGF inhibitor, DMH4 at a low dose to partially block ISA growth [54,55]. Similar to SU4516 treatment, wild type vessels have decreased in length following treatment with DMH4. Interestingly, although treatment with 15 μM or 25 μM DMH4 also reduces vessel length in *sema3fb* mutants, ISAs are still able to sprout an average 23 μm or 35 μm further than the treated wild type vessels, at each dose respectively (S6 Fig). Thus, slight reductions in Vegfr2 activity can restore *sema3fb* endothelial migration. Our data shows that the increase in *vegfr2* expression in *sema3fb* mutants has functional consequences on sprout migration.

## Discussion

Vegf activity must be tightly regulated during embryonic development and homeostasis [18,56–59]. This control is exquisite, and the zebrafish model has uncovered many regulatory layers of Vegfr2 pathway activity in endothelial cells [14,48,60]. Here we find that sema3fb offers an additional layer of control to VEGF signaling. Zebrafish *sema3fb* mRNA is expressed by developing trunk angioblasts. Loss of *sema3fb* stalls vessel migration and disrupts ISA morphology, resulting in wider and shorter ISAs in which nuclei are clumped and larger. Expression analysis revealed upregulation of *vegfr2* and *dll4* and a significant increase in pERK staining intensity in mutant endothelial cells. We also observed upregulation of an inhibitory molecule *svegfr1/sflt1* in mutant endothelium, suggesting *sema3fb* signaling also normally represses *sflt1* expression. Taken together, our results from an intact animal model suggest Sema3fb acts in an autocrine manner to regulate Vegf-mediated angiogenesis (Model; Fig 5).

In cultured endothelial cells, the anti-angiogenic effects of SEMA3F are primarily mediated through competitive binding of SEMA3F to NRP2 to antagonize VEGF-induced proliferation and migration, as observed in both cultured human umbilical vein endothelial cells (HUVECs) [61,62] and porcine aortic endothelial cells [63]. The SEMA3F-NRP pathway generally also opposes VEGF activity in tumor models, where a loss of SEMA3F leads to gross vessel overgrowth [29,63]. However, in our intact zebrafish *sema3fb* loss of function model, we were surprised to find reduced vessel growth despite upregulation of the *vegfr2* gene and an increase in

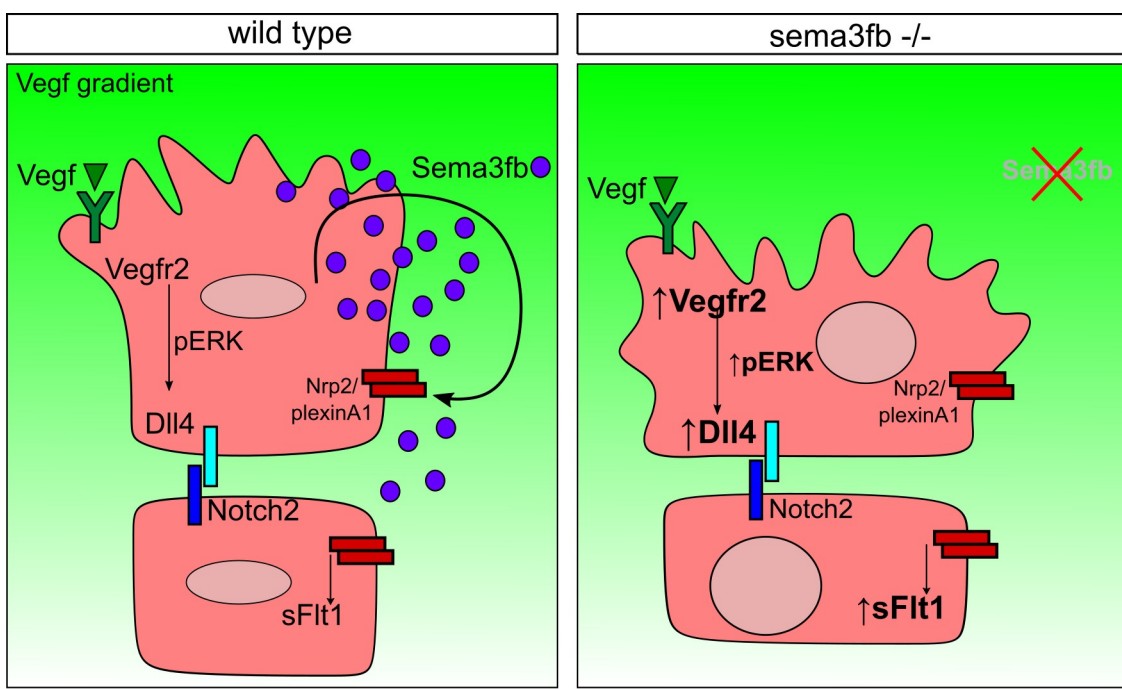

**Fig 5. Model of *sema3fb* action in sprouting vessels.** Sema3fb is expressed by endothelial cells during angiogenic sprout formation and acts through an autocrine mechanism to suppress Vegfr2 expression and maintain endothelial cell dynamics via controlling Dll4 expression and Notch signaling in addition to modulation of pERK. Loss of *sema3fb* increases Vegfr2, pERK, Dll4, and sFlt1 expression. This results in aberrant cellular morphology with wider sprouts, persistent filopodia, and larger nuclei. The changes in nuclear size and disrupted sprouting suggest a possible role for Sema3b to limit Vegf-mediated induction of downstream pERK signaling.

Vegfr2 activity. We find that inhibiting VEGF signaling rescued endothelial migration, suggesting that upregulation of Vegf signaling contributes to the observed phenotype. Intriguingly, the stunted sprout morphology of *sema3fb* mutants is also reflective of the phenotype reported for a gain of function for the inhibitory molecule *sflt1* [58]. Plexin-Semaphorin signaling is complex, with multiple co-expressed ligands and receptors present in cells that can mediate either additive or opposing effects. The Plexin receptor for Sema3f in zebrafish is not yet known, but in cell culture, Sema3F binds to PlexinA family receptors. PlexinD1 is a major endothelial cell receptor for other Sema3's (Sema3A, Sema3E) in mouse and has an exuberant angiogenic sprouting phenotype when lost. The fish *sema3fb* mutant phenotype appears opposite to that of *plexind1* loss of function [22,52]. Further, both *sema3fb* and *plexind1* regulate the expression of the decoy receptor *sflt1*, but in opposing, directions, suggesting their signaling has opposing action on common downstream pathways to fine-tune angiogenesis.

Increases in VEGF activity are reported to paradoxically increase both pro-angiogenic signaling via Dll4 as well as the expression of anti-angiogenic molecules such as sFlt1 in cell culture models [53]. Where excessive VEGFR-2 stimulation is reported to increase sFLT1 expression in HUVECs to limit inappropriate vessel growth [64,65]. If this feedback regulation is also at play in zebrafish, reducing excessive Vegfr2 activity in *sema3fb* mutants would be expected to decrease levels of sFlt1, thus allowing sprouting to occur. It is not obvious from the results of our expression analysis whether the stalled vessel phenotype occurs from an increase in Vegf signaling (upregulated *vegfr2* and pERK) or inhibited Vegf signaling (increased *sflt1*). Although we demonstrate that reducing Vegf signaling rescues the angioblast migration defect, we lack the tools to test protein levels of sFlt1, which might have distinct regulation, apart from

the mRNA levels we could measure here. The sFlt1 protein might contribute to our observed phenotype in ways that we cannot assess *in vivo*. For instance, the relative abundance of ligands (Sema3f), receptors (Nrp and Plexin), and co-receptor (Nrp2, note this is also a Vegfr2 co-receptor) present in a sprouting endothelial cell may influence intrinsic signaling responses. Increased *sflt1* mRNA may result in increased sFlt1 protein expression downstream of Vegfr2 upregulation, similar to what is observed in HUVEC cells [64]. However, increased mRNA may not result in increased protein, and protein subcellular localization may also play a role in the signaling. For instance, Vegf directly controls the expression of PlexinD1 to spatially restrict endothelial tip cell responses to Sema3e in retinal murine models [24]. Further, in cell culture models, there is higher SEMA3F expression on the leading edge of the cell that may influence directional tip cell migration of HUVECs [61,66].

Semaphorins traditionally play guidance roles to control vessel patterning. Interestingly, loss of Sema3fb does not affect spatial guidance during early angiogenesis. This either suggests that Sema3f signals through a different receptor or is one of several ligands of PlexinD1 in zebrafish, with other ligands conveying spatial guidance information. Indeed the synergy between Sema3A and Sema3F in HUVECs [36], suggests a combination of ligands more effectively antagonizes VEGF-mediated migration, and thus multiple Sema3s may control both the growth and trajectory of a sprout in the zebrafish trunk [67]. Studies in mouse and zebrafish demonstrate that inhibition of VEGFR2 results in dysregulation of actinomycin contractility necessary for endothelial cell elongation and vessel connections [68,69], similar to the phenotype we observe in *sema3fb* mutants. Sequestration of VEGF by SFLT1 further diminishes cell responses to limit endothelial cell size and actin rearrangements [70–72].

In cultured cells, exogenous SEMA3F controls F-actin/stress fiber formation, cell size, and elongation through disrupting the mTOR-RHOA/GTPase axis [30]. Although filopodia are dispensable for sprout initiation in zebrafish, assembly of F-actin into filopodia influences the speed, direction of migration, and anastomosis of endothelial cells into a connected vessel bed [73,74]. We observe that filopodia persist in *sema3fb* mutants (i.e., are abnormally stable). Loss of *sema3fb* may therefore hinder a pathway that destabilizes filopodia and vessel anastomosis may be impeded by underlying actin assembly defects. This is in line with the classic definition of Semas in regulating cell size and collapse of the cytoskeleton [11,75]. Our data applies to the role of Sema3fb in the initial formation and early elongation of ISA sprouts. In later development, RhoA- dependent mechanisms [76] control arterial lumen diameter and remodeling in response to changes in Vegf bioavailability and sFlt localization. Sema3fb may therefore have additional roles in regulating structural vessel adaptations; however, we were unable to assay late changes to vessel morphology in *sema3fb* mutants as they recover by this stage.

Here we demonstrate an autocrine role for Sema3f to modulate Vegf function *in vivo*. Together, our data suggest that Sema3fb acts to limit both positive (Vegfr2) and negative (sFlt1) signals, allowing endothelial cells to adopt an appropriate level of signaling downstream of Vegfr2 (Fig 5). Our data highlights a delicate balance of stop-go signals that can toggle the endothelial migration activity necessary for vessel growth. Together this work offers insight into understanding how context-dependent Sema3F modulation of endothelial responses to Vegf can be used to treat vascular disorders.

## Materials and methods

### Ethics statement

All animal procedures were approved by the University of Calgary Animal Care Committee (AC17-0189). Anesthesia and euthanasia used MS-222 (Tricaine) at 10–40 mg/L. Raw data underlying all graphs in the manuscript can be found in S1 Data.

## Zebrafish strains and maintenance

Wild type Tupfel longfin (TL) zebrafish or transgenic *Tg(-6.5kdrl:mCherry)*[ci5], *Tg(fli1a:nEGFP)*[y7] [50], *sema3fa*[ca304] [35], and s*ema3fb*[ca305] [43] were used for experiments. Embryos were collected within 10-minute intervals and incubated at 28.5˚C in E3 embryo medium (E3) and staged in hours post-fertilization (hpf). Endogenous pigmentation was inhibited from 24 hpf by the addition of 0.003% 1-phenyl-2-thiourea (PTU, Sigma-Aldrich, St. Louis, MO) to E3. To genotype, tissue was collected from single 30 hpf embryos following blinded imaging. Genomic DNA was extracted in 50μl of 50mM NaOH, boiled for 10 minutes and buffered with 1/10 volume of 100mM Tris-HCl pH 7.4, as described in [77] and amplified by PCR (as described by [43]) using the following primers: *sema3fb* forward (5'-ATTGCCCCACAAAAT AACATTC-3') and *sema3fb* reverse (5'-GTCTACTCTGTGAATTTCCCGC-3').

## Morpholino and *fli:lifeact* injections

Morpholinos (MO) against the *sema3fb* start codon (*sema3fb*[ATG]; ATG underlined) 5'- CATA-GACTGTCCAAGAGCATGGTGC-3' and against the *tnnt2a* start codon (tnnt2a[ATG], ATG underlined) 5'- CATGTTTGCTCTGATCTGACACGCA– 3' were obtained from Gene Tools LLC (Corvallis, OR, USA). Morpholinos were injected into one- to four-cell stage embryos within recommended dosage guidelines at 1ng/ embryo [78,79]. For endothelial-specific expression of Lifeact, a Tol2 construct using the Fli1[ep] promoter driving the expression of Lifeact-EGFP, *fli1*[ep]*Lifeact-EGFP* [73], was injected into one-cell stage embryos at 20 ng/μl plasmid with 25 ng/μl Tol2 transposase RNA.

## FAC sorting, RNA isolation, and RT-qPCR

For FACS analysis, 24 hpf *Tg(kdrl:mCherry)* wild type or mutant embryos were subjected to single-cell dissociation according to [80]. Briefly, embryos were washed once with calcium-free Ringers Solution and gently triturated 5 times before dissociation solution was added and incubated in a 28.5˚C water bath with shaking and periodic trituration for 20–30 min. The reaction was stopped, centrifuged, and resuspended in Dulbecco's Phosphate-Buffered Saline (GE Healthcare Life Sciences, Logan, Utah, USA), centrifuged and resuspended in fresh resuspension solution. The single-cell suspension was filtered with 75 μm, followed by 35 μm filters. Cells were sorted with a BD FACSAria III (BD Bioscience, San Jose, USA) and collected. Total RNA from 24hpf whole embryos or FACS sorted cells was isolated using the miRNeasy Mini Kit (Qiagen). 500 ng of total RNA from each sample was reverse transcribed into cDNA using SuperScript III First-Strand Synthesis SuperMix (18080–400; Invitrogen). cDNA in a 5ng/10ul final reaction was used in a TaqMan Fast Advanced Master Mix (Thermo Fisher Scientific, Massachusetts, U.S). Reactions were assayed using a QuantStudio6 Real-time system (Thermo Fisher Scientific). Zebrafish specific Taqman assays (Thermo Fisher Scientific, Waltham, Massachusetts, USA) were used: vegfab (Cat# 4448892, Clone ID: Dr03072613_m1), kdrl(vegfr2) (4448892, Dr03432904_m1), dll4 (4448892,Dr03428646_m1), notch2 (4448892, Dr03436779_m1), jag1a (4448892, Dr03093490_m1), sflt1 (4331448, ADP47YD4) and normalized to β-actin (4448489, Dr03432610_m1). The ΔΔCt method was used to calculate the normalized relative expression level of a target gene from triplicate measurements. Experiments were repeated three times independently unless stated otherwise.

## Drug treatments

For Vegfr2 inhibition, a 200 μM SU5416 (Sigma #S8442) stock solution in DMSO (Sigma #D8418) or a 100 μM DMH4 (Sigma #8696) stock solution in DMSO was diluted to working

concentrations in fish water. For SU5416 final concentrations of 0.2 μM SU5416 and 0.5 μM in 0.1% DMSO were used as described in [53,56,81]. For DMH4 15 μM and 25 μM in 0.1% DMSO was used as described in [54,55]. Embryos were treated from 20 to 30 hpf to target both ISA and DLAV angiogenesis. WT and homozygous *sema3fb*[ca305] mutant embryos were manually dechorionated before receiving the SU5416 or DMH4 and control (0.1% DMSO) treatments using a common solution for both genotypes.

## In situ hybridization and immunostaining

All embryos were fixed in 4% paraformaldehyde in PBS with 0.1% Tween-20 at 4˚C overnight, followed by 100% methanol at −20˚C.

For colorimetric in situ hybridization (ISH) Digoxigenin (DIG)-labeled antisense RNA probes were synthesized using SP6 or T7 RNA polymerase (Roche, Basel, Switzerland). Digoxigenin RNA probes were generated from plasmid template (*sema3fa*—pCR4, linearized with PmeI) (*sema3fb*—pCR4, linearized with NotI). Wholemount and section in situ hybridization were performed as described by [82] with some minor modifications: steps with gradient changes in hybridization buffer: 2 x SSC: and 0.2 x SSC:PBST were carried out at 70˚C and 0.2 x SSC at 37˚C. NBT/BCIP was used at a 2.5/3.5μl/ml ratio, respectively. Stained embryos were transferred to microcentrifuge tubes, fixed in 4% paraformaldehyde (PFA), and washed in 1 x PBST before imaging. For transverse sections of the trunk, following wholemount ISH embryos were embedded in JB4 medium (PolySciences, Warrington, PA). Briefly, embryos were fixed in 4% PFA overnight at 4˚C, washed in PBS, and dehydrated in 100% EtOH. Following dehydration, embryos were soaked in infiltration solution (2-hydroxyethyl methacrylate/benzoyl peroxide) until they sank to the bottom of the tube and then transferred and positioned in molds filled with embedding solution (infiltration with N-Dimethylaniline), allowed to harden overnight, and sectioned at 7 μm using a Leica microtome.

For Hybridization Chain Reaction (HCR) in situ hybridization, custom probes were obtained from Molecular Instruments (Los Angeles, CA) for soluble Flt1 (targeted to the unique region of the gene) and VEGFR2. The in-situ staining reactions were as recommended by the manufacturer.

For antibody staining, embryos were permeabilized in 50:50 acetone/methanol for 20 minutes, rehydrated at room temperature, and then blocked in 10% normal sheep serum in PBST with 1% triton, and incubated for at least 48 hours at 4˚C in primary antibody. Immunostaining for Laminin (1:400, Sigma-Aldrich, Missouri, United States), mCherry (1:200, Developmental Studies Hybridoma Bank, Iowa City, United States), and phosphoERK (pERK) (1:500, Cell Signaling 43701:Danvers, MA, USA) were detected with mouse anti-mCherry antibody, (1:500, Clontech, Mountain View, California, USA) and detected with Alexafluor 555 or 488 secondary antibodies (1:500; Invitrogen), for 1 hour at RT in 5% NSS/PBST/0.1% triton.

## Image acquisition and measurements

For wholemount imaging, embryos were immobilized in 0.004% Tricaine (Sigma) and mounted in 0.8% low melt agarose on glass-bottom dishes (MatTek, Ashland, MA). Confocal images were collected on a Zeiss LSM 700 inverted microscope. Slices 1–3 μm apart were gathered and processed in Zen Blue 2012 software and presented as maximal intensity projections (z-stacks, total of 10–15 slices per embryo/image), and analyzed using FIJI/ImageJ [83].

For time-lapse, z-stacks were acquired every 10–30 min for 5 hours, from 25 hpf—29 hpf. ZEN Blue 2012 software was used to extract timepoints and image processing/measurements were performed using Fiji (ImageJ).

Unless otherwise, ISA measurements were performed on at least 8 ISAs per embryo. ISA length measurements were made with a segmented line tool along the vessel from the edge of the aorta to the leading edge of the spout. To calculate the % connected, each ISA connection to its neighboring sprout was counted and expressed as a percentage of the total number of ISA counted. For ISA sprout diameter, 5 sprouts were measured at the boundary of the horizontal myoseptum. For DA and PCV measurements, 5 measurements were taken along the DA and PCV, directly below the ISAs above the yolk extension. For endothelial cell nuclei spacing, 2 measurements were taken from the middle of each nucleus to the next nucleus. To measure fluorescent kdrl:mCherry intensity, total fluorescence (TF) was calculated using the formula: TF = Integrated Density—(Area of selected cell X Mean fluorescence of background readings). The area for measurement was gated by tracing 5 ISVs above the yolk extension.

For angioblast migration and ISA speed calculations, analysis was from images obtained at 30min intervals from 25 hpf– 29hpf. For ISA length over time, 5 ISAs were measured per embryo (as described above) at each extracted timepoint. For ISA migration speed, changes in ISA length were compared at 1hr intervals, 25 (T1)– 26(T2) hpf, 26 (T2)- 27(T3) hpf, 27(T3)– 28(T4) hpf, and 28 (T4)– 29 (T5) hpf, and distance traveled was calculated using the formula Distance = End Length (μm)–Start Length (μm). (e.g., Distance from 25–26 hpf = Length at T2 –Length at T1). The speed of ISA migration at each interval was calculated using the formula Speed = Distance (μm) /Time (min) (See S1 and S2 Tables) For lead angioblast migration, the distance of a fli1a:nEGFP positive nucleus from the DA was measured at 1-hour intervals from 25–29 hpf.

## Statistics

All data sets for quantitation (qualitative scorings or absolute measurements) were analyzed blinded. Results are expressed as mean ± SD. All statistical analysis was performed using Prism 7 software (Graph Pad). Unpaired, nonparametric tests were used for all statistical tests, either the Student T-test with Welch's correction for comparisons of two samples, or Two-Way ANOVA with a Kruskal-Wallis test for comparisons between multiple samples.

## Supporting information

**S1 Fig. *sema3f* expression and *sema3fa* mutant phenotype.** A) Lateral view of whole-mount ISH from 26-30hpf shows *sema3fa* expression in the ventral lateral somites and *sema3fb* expression in the dorsal aorta (DA) and intersegmental arteries (ISAs). HM: Horizontal Myoseptum. B) Expression of *sema3fa* and *sema3fb* in transverse sections of the trunk at 28hpf. *sema3fa* is expressed in ventral and lateral somite (arrows) while sema3fb is strongly expressed in the DA (arrowhead), Neural tube (nt), notochord (nc). C) Lateral confocal images of trunk vasculature (black) of 30hpf control wild type (WT), homozygous *sema3fa*ca304 and *sema3fb*ca305 mutant embryos with and without injection of 1ng *sema3fb*ATG-MO. Scale bar, 100 μm. n/N = number of embryos with angiogenic defects/Total number of embryos. D) Length of ISA sprouts in WT, *sema3fb*ATG-MO morphants and *sema3fb* ca305 knockdown embryos with *sema3fb*ATG-MO at 30 hpf; N = 1, WT = 2 embryos (10 ISAs, mean 105±7 μm), *sema3fb*ATG-MO = 5 embryos (25 ISAs, mean of 78±20 μm), and *sema3fb*ca305 + *sema3fb*ATG-MO 5 embryos (25 ISAs, mean 87±15 μm), ****p<0.0001. E) Length of ISA sprouts in WT and *sema3fa*ca304 mutant and embryos with *sema3fb*ATG-MO at 30 hpf; N = 2, 6 embryos per group: WT (30 ISAs, mean 109±7 μm), *sema3fa*ca304 (28 ISAs, mean of 106±9 μm), and *sema3fa*ca304 + *sema3fb*ATG-MO (30 ISAs, mean 91±17 μm), ****p<0.0001. One-Way ANOVA Tukey's multiple comparisons test. Error bars = ±SD. (TIF)

**S2 Fig. Loss of *sema3fb* angiogenic deficits are independent of blood flow.** A) Confocal lateral images of laminin-stained embryos at 30hpf. Tg(kdrl:mCherry) endothelium (red) and laminin (green). Embryos derived from a heterozygous *sema3fb*$^{ca305/+}$ incross. B) Quantification of the length of ISA sprouts at 30hpf, N = 1: wild type (WT) (40 ISAs, 4 embryos, mean of 102±1 μm$^2$), *sema3fb*$^{ca305/+}$ (120 ISAs, 12 embryos, mean of 90±8 μm$^2$), and *sema3fb*$^{ca305}$ (150 ISAs, 15 embryos, mean of 90±14 μm$^2$). C) Confocal lateral images of the trunk endothelium (black) in blood flow-stopped *tnnt2*ATG-MO injected wild type siblings (WT) and *sema3fb*$^{ca305}$ mutants. DLAV gaps (blue asterisks) and truncated ISAs sprouts (yellow arrowheads) are marked. Scale bar, 100μm. D) Length of ISA sprouts at 30 hpf, N = 3: WT (80 ISAs, 8 embryos, mean length of 106±3 μm), WT + *tnnt2a*$^{ATG-MO}$ (100 ISAs, 10 embryos, mean 98 ±8.4 μm), *sema3fb*$^{ca305}$ (80 ISAs, 8 embryos, mean 86±17 μm), and *sema3fb*$^{ca305}$ + *tnnt2a*$^{ATG-MO}$ (100 ISAs, 10 embryos, mean 85±22 μm). E) Percentage of ISAs connected at DLAV at 30 hpf, N = 3: WT (mean 82±9% connected), WT + *tnnt2a*$^{ATG-MO}$ (mean 86±7% connected), *sema3fb*$^{ca305}$ (80 ISAs, 8 embryos, mean 54±15% connected), and *sema3fb*$^{ca305}$ + *tnnt2a*$^{ATG-MO}$ (mean 54±16% connected). F) Quantification of width of DA in 30 hpf embryos, N = 3: WT (8 embryos, 5 measurements per embryo/n = 40 total, mean width of 18 ±3 μm), WT + *tnnt2a*ATG-MO (10 embryos, 5 measurements per embryo/n = 50 total, mean 9±1 μm), *sema3fb*$^{ca305}$ (8 embryos, 5 measurements per embryo/n = 40 total, mean 11±3 μm), and *sema3fb*$^{ca305}$ + *tnnt2a*$^{ATG-MO}$ (10 embryos, 5 measurements per embryo/n = 50 total, mean 10±1 μm). G) Quantification of width of PCV in 30 hpf embryos, N = 3: WT (n = 40, mean width of 22±3 μm), WT + *tnnt2a*$^{ATG-MO}$ (n = 50, mean 19±4 μm), *sema3fb*$^{ca305}$ (n = 40, mean 20±3 μm), and *sema3fb*$^{ca305}$ + *tnnt2a*$^{ATG-MO}$(n = 50, mean 19±4 μm). 2-Way ANOVA Tukey's multiple comparisons test, **** means p = <0.001. Error bars = ±SD. (TIF)

**S3 Fig. Sema6b mutants recover by 48 hpf.** A) Representative Lateral confocal images of trunk vasculature (black) of 48 hpf control wild type (WT), heterozygous *sema3fb*$^{ca305/+}$ and homozygous *sema3fb*$^{ca305}$ mutant embryos with no obvious differences in vessel morphology or Segmental vessel (Se) connections. Scale bar, 100 μm. SIV = sub-intestinal vein (plexus), DA = Dorsal Aorta, PCV = Post Caudal Vein, CA = Caudal Artery, CV = Caudal Vein. N = 2, n: WT = 5, *sema3fa*$^{ca305/+}$ = 5, and *sema3fb*$^{ca305}$ = 7. (TIF)

**S4 Fig. *sema3fb* morphant endothelial nuclei phenotype.** A) Lateral confocal timelapse images of time-lapse images in 25.5–28.5 hpf double transgenic Tg(kdrl:mCherry;fli1a: nEGFP) endothelial cells (magenta) and nuclei (white). The horizontal myoseptum (green dashed line) is noted to highlight ISA growth over time. Scale bar, 50 μm. B) During the 25–26 hpf interval there is no significant difference in speed between wild type and mutant embryos, see S2 Table for details. C) Lead angioblast mean distance from DA at 25 hpf: WT = 44.47 ±8.36 μm and *sema3fb*$^{ca305}$ = 43.14±6.84 μm, p = 0.609. D) Lead angioblast at 26 hpf mean distance from DA: WT = 55.12±14.06 μm and *sema3fb*$^{ca305}$ = 47.18±5.75 μm, p = 0.572. C-D) N = 1: WT = 4 embryos (20 ISAs) and *sema3fb*$^{ca305}$ = 3 embryos (15 ISAs), Unpaired t-test with Welch's correction. E) Lateral confocal images of double transgenic Tg(kdrl:mCherry; fli1a:nEGFP) endothelium (red) and endothelial cell nuclei (green). DLAV gaps (blue asterisks) and truncated ISAs sprouts (white arrowheads) are noted. Embryos derived from a heterozygous *sema3fb*$^{ca305/+}$ incross. Scale bar, 100 μm. G) Quantification of the number of endothelial cell nuclei per ISAs in 30 hpf embryos, N = 2: WT (6 embryos, mean of 3 nuclei/ ISA)), and *sema3fb*MO (7 embryos, mean of 3 nuclei /ISA). Unpaired t-test with Welch's correction, p = 0.17. G) Quantification of the average area of endothelial cell nuclei per ISAs in 30

hpf embryos, N = 3: WT (60 ISAs, 6 embryos, mean of 42±16 μm$^2$), and *sema3fb*MO (70 ISAs, 7 embryos, mean 61±19 μm$^2$). Unpaired t-test with Welch's correction, ****p<0.0001. Error bars = ±SD

(TIF)

**S5 Fig. *sema3fb* mutants display aberrant and persistent filopodia.** A) Representative still images of single-cell expression of fli1ep: Lifeact-EGFP (green) in ISA endothelial cells from 28-30hpf wildtype and *sema3fb*$^{ca305}$ embryo time-lapse imaging. A dashed white line represents the horizontal myoseptum and selected areas for filopodia counts are highlighted in white boxes. B) Quantification of number Lifeact-EGFP positive filopodia on ISA at 28hpf from embryos of the indicated genotypes. Unpaired t-test, p = 0.3566. C) Quantification of number Lifeact-EGFP positive filopodia on ISA at 29hpf from embryos of the indicated genotypes. Unpaired t-test, p = 0.0029. D) Quantification of number Lifeact-EGFP positive filopodia on ISA at 30hpf from embryos of the indicated genotypes. N = 3 for each quantification: WT (14 ISAs, 6 embryos, mean of 3 filopodia/ISA) and homozygous sema3fb$^{ca305}$ (18 ISAs, 7 embryos, mean of 8 filopodia/ISA). Unpaired t-test, p = 0.0002.

(TIF)

**S6 Fig. Interactions between VEGFR inhibitor and sprouting in *sema3fb* mutants.** A) A model of signaling pathways that regulate angiogenic sprouting, highlighting key genes controlling tip and stalk cell identity. B) Quantification of Tg(kdrl:mCherry) transgene expression levels in wild type and *sema3fb*$^{ca305}$ ISAs at 30hpf. N = 3: wild type (n = 13 embryos, average of 6500 a.u.) and *sema3fb*$^{ca305}$ (n = 14 embryos, average of 8400 a.u.). T-test with Welches correction, *p = 0.0186, a.u. = arbitrary unit of intensity. C). B) Fluorescent HCR in situ for vegfr2 and *sflt1* RNA transcripts in whole-mount wild type and embryos *sema3fb*$^{ca305}$ embryos fixed at 30 hpf. D) Whole-mount Immunostaining for phosphoERK (pERK) in WT and *sema3fb*$^{ca305}$ embryos fixed at 30 hpf. E) Lateral confocal images of the trunk vasculature Tg (kdrl:mCherry) (white) in embryos treated with 0.5 μM SU5416 from 20hpf-30hpf. DLAV gaps (blue asterisks) and ISA truncated sprouts (yellow dashed line at the level of horizontal myoseptum are indicated. Scale bar, 100 μm. F) Quantification of ISA sprout length in 30 hpf embryos treated with 0.5 μM SU5416, N = 1: WT + DMSO (25 ISAs, 5 embryos, mean of 107 ±8 μm), WT + 0.5μM SU5416 (25 ISAs, 5 embryos, mean of 50±14 μm), *sema3fb*$^{ca305}$ + DMSO (30 ISAs, 6 embryos, mean of 82±17 μm), and *sema3fb*$^{ca305}$ +0.5μM SU5416 (30 ISAs, 6 embryos, mean of 82±19 μm). G) Percentage of ISA sprouts connected at DLAV in 30 hpf embryos treated with 0.5 μM SU5416, N = 1: WT + DMSO (25 ISA-DLAV, 5 embryos, mean 78% of ISA-DLAV/embryo), WT + 0.5 μM SU5416 (25 ISA-DLAV, 5 embryos, mean of 78%), *sema3fb*$^{ca305}$ + DMSO (30 ISA-DLAV, 6 embryos, mean of 51%), and *sema3fb*$^{ca305}$ + 0.5 μM SU5416 (30 ISA-DLAV, 6 embryos, mean of 82±19%). Error bars = ±SD. H) Lateral confocal images of the trunk vasculature Tg(kdrl:mCherry) (white) in embryos treated with low doses of DMH4 μM SU5416 from 20hpf-30hpf. ISA truncated sprouts (yellow dashed line at the level of horizontal myoseptum are indicated. Scale bar, 50 μm. I) Quantification of length of ISA sprouts in 30 hpf embryos treated with 15 μM DMH4, J) Quantification of length of ISA sprouts in 30 hpf embryos treated with 25 μM DMH4. I-J) N = 1; WT+ DMSO n = 3, 15 ISAs, average (ave.) 102±9 μm; WT+ μM DMH4 n = 2, 10 ISA, ave. 17±17 μm; WT+ 25 μM DMH4 n = 3, 15 ISAs, ave. 7±8 μm. sema3fb$^{ca305}$ + DMSO = 3, 15 ISA, ave. 87±18 μm, sema3fb$^{ca305}$ + 15 μM DMH4 n = 2, 10 ISA, ave. 40±16 μm; sema3fb$^{ca305}$ + 25 μM DMH4 n = 3,15 ISA, 32 ±13 μm. One-Way ANOVA Tukey's multiple comparisons test, * means ¬¬p = 0.012, *****p = <0.0001. Error bars = ±SD.

(TIF)

**S1 Table. ISA length at different time intervals.**
(DOCX)

**S2 Table. Angioblast migration distance and speed.**
(DOCX)

**S3 Table. Quantitative PCR mean fold change data.**
(DOCX)

**S1 Data. Raw data underlying all graphs in the manuscript.**
(XLSX)

## Acknowledgments

We thank Li Kun Phng for providing us with the FliLifeact:GFP construct. We thank the University of Calgary FACS core for assistance with cell sorting. We thank Suchit Ahuja, Jasper Greysson-Wong, and Jae-Ryeon Ryu for comments on the manuscript, and Carrie Hehr for sema3fb^ATG-MO morpholino injections.

## Author Contributions

**Conceptualization:** Charlene Watterston, Rami Halabi.

**Formal analysis:** Charlene Watterston.

**Funding acquisition:** Sarah McFarlane.

**Investigation:** Charlene Watterston, Sarah J. Childs.

**Methodology:** Charlene Watterston, Rami Halabi, Sarah J. Childs.

**Project administration:** Sarah J. Childs.

**Resources:** Rami Halabi, Sarah J. Childs.

**Supervision:** Sarah McFarlane, Sarah J. Childs.

**Visualization:** Charlene Watterston.

**Writing – original draft:** Charlene Watterston.

**Writing – review & editing:** Charlene Watterston, Sarah McFarlane, Sarah J. Childs.

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
