## [Decision Letter · Decision Letter 0]

19 Mar 2021

Dear Dr Sarah,

Thank you very much for submitting your Research Article entitled 'Endothelial Semaphorin 3fb regulates Vegf pathway-mediated angiogenic sprouting' to PLOS Genetics. I apologize for the delay in returning this decision.

The manuscript was fully evaluated at the editorial level and by twi independent peer reviewers. The reviewers agreed that the study was well performed and novel, but raised some substantial concerns about the current manuscript. They would like you to consider alternative explanations for some of your findings, and for you to validate your model with further experiments. Based on the reviews, we will not be able to accept this version of the manuscript, but we would be willing to review a much-revised version. We cannot, of course, promise publication at that time.

If you decide to revise the manuscript for further consideration at PLOS Genetics, please aim to resubmit within the next 60 days, unless it will take extra time to address the concerns of the reviewers, in which case we would appreciate an expected resubmission date by email to plosgenetics@plos.org.

[LINK]

We are sorry that we cannot be more positive about your manuscript at this stage. Please do not hesitate to contact us if you have any concerns or questions.

Yours sincerely,

Cecilia Moens

Associate Editor

PLOS Genetics

Gregory P. Copenhaver

Editor-in-Chief

PLOS Genetics

Reviewer's Responses to Questions

**Comments to the Authors:**

Reviewer #1: In this manuscript Watterston and colleagues investigate the function of the sema3fb gene in zebrafish vascular development. Using a sema3fb mutant that is predicted to express a truncated transcript, they reveal that intersegmental artery elongation is stunted and filopodia persist for longer in heterozygous and homozygous mutant embryos. These observations are associated with an increase in gene expression of angiogenesis inducing and repressing factors. Finally, Watterston et. al. nicely demonstrates that disruption intersegmental artery elongation in sema3fb mutants is reversed by 0.2µM SU5416, suggesting that excessive Vegfr2 activity underpins this mutant defect. The study is well presented and I am positive about this work, however there are number of remaining important questions that need to be addressed before publication.

(1) Is Vegfr2 signaling truly increased in sema3fb mutants?

At several points in the text it is suggested that the observed phenotype is due to increased vegfr2 expression resulting in an assumed increase in Vegfr2 activity in sema3fb mutants. However, increased vegfr2 expression does not confirm increased Vegfr2 activity, especially when sflt1 expression is also increased. The suggestion of increased Vegfr2 activity in sema3fb mutants needs to be confirmed to support the author’s model. Others have used pErk immunostaining or Erk reporters to characterize Vegfr2 activity during intersegmental artery elongation. These would be important tools to use to confirm the author’s hypothesis that increased vegfr2 expression leads to increased Vegfr2 activity.

(2) Could defects in vessel branching be due to a delay in sema3fb mutant development?

Many aspects of the observed phenotype of sema3fb heterozygous and homozygous mutant embryos could be due to an embryonic delay or transient disruption to vessel branching. For example, does vessel elongation fully recover at later time points? Importantly, can the authors confirm that sema3fb mutant do not experience an overall developmental delay by quantifying classic embryonic stage markers, such as head-trunk angle and embryo length.

(3) Are persistent filopodia in sema3fb mutant a consequence of delayed anastomosis?

The authors reveal that sema3fb mutants retain intersegmental artery filopodia for longer than wild type siblings. However, earlier in the manuscript they also show that sema3fb mutant intersegmental arteries often fail to connect with each other by anastomosis to form the DLAV. As this anastomosis event is responsible for repressing filopodia, it seems possible that the persistent filopodia observed in sema3fb mutants could be an indirect consequence of this delayed anastomosis rather than a specific effect of sema3fb on actin remodeling. To clarify this point, the authors should confirm that they only quantified filopodia numbers on intersegmental arteries in sema3fb mutants that connected to form the DLAV at the exact same time point as in wild type sibling embryos (and not in slightly delayed vessels). If not, it would be important to reanalyze the data to take this into account.

(4) Is endothelial cell motility disrupted in sema3fb mutants?

The short intersegmental arteries observed are likely a consequence of reduced endothelial cell motility, but this is not tested. Likewise, changes to endothelial cell nuclei spacing and shape are also potentially a consequence of reduced endothelial cell movement. It would be important to image intersegmental artery elongation in live embryos and quantify endothelial cell motility to confirm the cause of these vessel defects.

(5) Does sema3fb truly form part of a “feedback mechanism”?

At several points in the text the authors indicate that sema3fb forms part of a feedback mechanism that regulates Vegfr activity. This concept is interesting, but is not directly tested. The authors should determine if sema3fb gene expression is regulated by Vegfr signaling using mutants or chemical inhibitors of the Vegf pathway. If Vegf induces sema3fb expression, which then feeds back to inhibit excessive Vegfr activity, this might provide evidence of a bona fide feedback loop.

(6) What is the role of sflt1 in sema3fb-mediated control of intersegmental artery elongation?

On line 286, the authors propose that “reducing excessive Vegfr2 activity in sema3fb mutants would be expected to decrease levels of sFlt1, thus allowing sprouting to occur”. This is somewhat counterintuitive as high Vegfr2 activity is considered to induce sprouting. Moreover, it is unclear in this model how endothelial cells can maintain high levels of Vegfr2 activity if sFlt1 levels are so high that they repress this Vegfr2 activity. To support this proposed model, the authors should determine by qPCR if sflt1 levels return to normal in sema3fb mutants upon treatment with 0.2µM SU5416 or other Vegfr2 inhibitors.

Minor points:

- It would be important to confirm the specificity of the sema3fb morpholino tool (as well as sema3fb haploinsufficiency) by injecting this morpholino into heterozygous and homozygous sema3fb mutant embryos to confirm that no additional phenotype is observed.

- SU5416 is not a specific inhibitor of Vegfr2 signaling and impacts many unrelated tyrosine kinases. The authors should confirm the important findings in Fig 4 using a more specific inhibitor of Vegfr2 (or if available, kdr or kdrl heterozygous and homozygous mutant embryos).

- Lines 314 to 319. The discussion (and Fig 5) introduces the concept that PI3K signaling downstream of VEGFR2 may be disrupted in sema3fb mutants. However, this statement is unclear and the case for an involvement of PI3K is weak. For example, how would this account for the increased vegfr2, dll4 and sflt1 levels seen in sema3fb mutants? Fig 5 would benefit from removal of PI3K from the illustration, as this is confusing, and a clearer case for PI3K involvement should be added to the discussion (or this section removed altogether).

Reviewer #2: The manuscript by Watterston et al focuses on the role of Semaphorin 3fb in vascular development in zebrafish embryos. The authors have made zebrafish mutants in sema3fa and sema3fb genes. Sema3fb (and not sema3fa) show vascular sprouting defects and misregulated expression of genes associated with VEGF signaling. Remarkably, reduced ISV sprouting is rescued by low dose treatment of VEGF chemical inhibitor. The authors conclude that Sema3fb functions by modulating VEGF pathway in angiogenic sprouting.

Overall, the study is well performed, the images are clear, the results are novel and describe a new role for a zebrafish Sema3F homolog in promoting angiogenesis. However, the analysis of the phenotype is mostly descriptive and leaves many questions unanswered, such as, how loss of Sema3fb expression results in increased kdrl and flt1.

Specific points:

1. There are multiple references to the doctoral dissertation by Halabi R. For example, data regarding the sema3fb mutant generation (which specific nucleotides are deleted in the gene), or the presence of cardiac defect are not shown and the reference by Halabi R is cited instead. It is not sufficient to cite data reported in the doctoral dissertation because it is not considered a peer-reviewed publication. Any data that is relevant to the manuscript and has not been published in a peer-reviewed journal should be included in this manuscript so that it can be reviewed.

2. Similarly, it is said that sema3fb morpholino has been validated and the dissertation by Halabi R is cited. All data for morpholino validation need to be shown in this manuscript. Does MO cause a similar inhibition of ISV sprouting and DLAV connections as observed in sema3fb mutant?

3. More complete description of the phenotype observed in sema3fb mutants is needed. Do sprouting defects recover at later stages? Are there any other vascular defects observed? Are there any obvious morphological defects observed? Cardiac defect has been mentioned; please include a description. Are homozygous mutants viable?

4. It is important to include analysis of sema3fb MO embryos alone in Suppl. Fig. 1D and E to shown that the phenotype in double inhibited embryos is the same as in sema3fb mutants / morphants alone.

5. The text says that sema3fb fish show a slight reduction in both the DA and PCV width. Yet there is no statistically significant difference in PCV width between wt and sema3fb mutants (Suppl. Fig. 2G). Also, please explain how DA and PCV were measured; is it a single measurement per embryo? It says N=3 in the figure legend for Suppl. Fig. 2F and 2G; does it mean 3 embryos; there are multiple data points in the figure? Please check the reference to panel F in the figure legend.

6. It would be very informative to perform fluorescent in situ (or RNA-Scope) for kdrl, dll4 and sflt1 in sema3fb mutants. qPCR data cannot show differences in spatial localization, and these transcripts are supposed to be expressed differently in tip and stalk cells. This would allow to assay which cells (tip, stalk) show differences in transcript expression.

7. SU5416 is not specific to vegfr2, and can inhibit other tyrosine kinase receptors including vegfr3 / flt4. It would be beneficial to confirm the SU5416 rescue experiment using genetic approaches that reduce VEGF signaling. Would ISV sprouting be restored in sema3fb mutants crossed into kdrl mutant background (or kdrl morpholino can be used instead)?

8. Because opposing results (increased kdrl and sflt1 expression) are observed it is unclear how VEGF signaling is affected in forming sprouts. Direct readout of VEGF signaling can be measured by quantifying pERK expression (from immunostaining or a live reporter line) which would help to confirm changes in VEGF signaling in sema3fb mutants.

**Have all data underlying the figures and results presented in the manuscript been provided?**

Reviewer #1: Yes

Reviewer #2: **No: **There are multiple references to the doctoral dissertation by Halabi R. For example, data regarding the sema3fb mutant generation (which specific nucleotides are deleted in the gene), or the presence of cardiac defect are not shown and the reference by Halabi R is cited instead. It is not sufficient to cite data reported in the doctoral dissertation because it is not considered a peer-reviewed publication.

PLOS authors have the option to publish the peer review history of their article (what does this mean?). If published, this will include your full peer review and any attached files.

Reviewer #1: No

Reviewer #2: No

---

## [Decision Letter · Decision Letter 1]

26 Jul 2021

Dear Sarah,

Thank you very much for submitting your Research Article entitled 'Endothelial Semaphorin 3fb regulates Vegf pathway-mediated angiogenic sprouting' to PLOS Genetics.

The manuscript was fully evaluated at the editorial level and by independent peer reviewers. Two reviewers who reviewed the first version of the manuscript appreciate your effort to address their criticisms and now recommend acceptance pending minor revisions that you can easily make. However the third reviewer, who saw the manuscript for the first time, had more significant concerns. They would like to see a better effort to address what they see as contradictions between your work and the existing literature. They also ask for additional experiments. I propose that you do the former but dispense with the further experiments that they propose unless you feel that they would strengthen this already-strong manuscript.  

[LINK]

Yours sincerely,

Cecilia Moens

Associate Editor

PLOS Genetics

Gregory P. Copenhaver

Editor-in-Chief

PLOS Genetics

Reviewer's Responses to Questions

**Comments to the Authors:**

Reviewer #1: The authors have addressed almost all of my comments appropriately with new experimental data or text clarifications. I have just one point that I would like clarified prior to acceptance for publication:

New data addressing the impact of sema3fb mutation on pERK levels (Fig. 4F) indicates that only homozygous null embryos exhibit an increase in pERK, whereas heterozygous embryos are unaffected. However, this is not consistent with other presented data that indicates that sema3fb is haploinsufficient. A full discussion of this discrepancy and a potential explanation is required prior to publication.

Reviewer #2: In the revision authors have satisfactorily addressed all of my major concerns. There are only a few minor issues that still need to be addressed.

1. It would be very helpful to refer to specific panels when referencing supplemental figures. Currently in most cases the reference is made to the entire figure (such as S2 Fig.) without referencing individual panels.

2. Supplemental figures show in either gray background or no background (depending which application is used to open them) which can be distracting and makes it harder to see the text. I would strongly suggest inserting white background and saving all supplemental files as .pdf (instead of .png) to make them more compatible to different readers.

3. It would be helpful to include supplemental table legends next to each table (not in a separate document).

4. Text on page 9 says that DMH4 at low dose rescued mutant vessel length. This statement is inaccurate because addition of DMH4 in sema3fb mutants resulted in shorter sprouts than sema3fb + DMSO embryos, thus addition of DMH4 made the phenotype more pronounced. This statement should be revised.

5. Test should be checked carefully for grammar and errors. There are several errors throughout the text (‘23µ’ should be ‘23 µm’ on page 9, ‘ha’ should be ‘has’ on page 11, ‘find there is a significant increase fluorescent intensity’ should be ‘find that there is a significant increase in fluorescence intensity’ on page 9).

6. S6 Fig. is described in the Discussion section. However, it shows original experimental results that were not discussed previously, therefore it would be more appropriate to move it to the Results section and describe it there.

Reviewer #3: Comments to authors:

This manuscript is conveying a “confusing message”. On the one hand the authors propose that the loss of Sema3fb augments Vegf signaling. On the other hand, the authors provide evidence for upregulation of sFlt1 (Vegf-trap), which would argue for reduced Vegf bio-availability and kdrl signaling.

In my view the phenotype observed upon loss of sema3fb is subtle, a slight reduction (or delay) in the expansion of developing arterial ISVs. The described phenotype is not necessarily a phenocopy of Vegf loss - or gain of function. Zebrafish embryos with increased vegfaa expression show normal aISV development and filopodia numbers on aISV tip cells (Wild et al, Nat Comms, 2017). Conversely, sflt1 gain of function results in truly shorter aISVs, that in general do not grow beyond the horizontal myoseptum. Therefore, in my opinion, the data with the R2 receptor inhibitor restoring aISV progression are puzzling. I agree with reviewer 2 that the authors should try low dose vegfaa targeting morpholino of examine vegfaa+/- or kdrl+/- heterozygous mutants instead of the pharmacology tools. In the same line, the authors could examine the impact of reducing sflt1 expression levels (using low dose flt1 targeting morpholino of flt1+/- mutant) on vascular morphology, in the sema3fb mutant. The claim that increased Vegf signaling augments sflt1 expression can be substantiated in other appropriate vegf – kdrl gain of function settings (like vhl-/- mutants). The authors may want to consider the TgBAC(flt1:YFP) reporter to study the flt1 expression patterns.

The authors try to link changes in endothelial nuclear cell morphology with vessel width. It has recently been shown that endothelial R2 signaling promotes endothelial cell enlargement, resulting in aISVs with a larger lumen diameter in zebrafish embryos (Klems, Nat Comms 2020). As the authors propose increased endothelial R2 signaling in sema3fb mutants, the authors should check changes in endothelial cell shape as an explanation for changes in vessel width.

The authors speculate that Sema3fb may control F-actin and cell elongation through disruption of the RhoA/GTPase pathway (based on reference 30). The authors can test this in their scenario – there are appropriate small Rho-GTPase inhibitors that can be used in zebrafish.

**Have all data underlying the figures and results presented in the manuscript been provided?**

Reviewer #1: Yes

Reviewer #2: Yes

Reviewer #3: Yes

PLOS authors have the option to publish the peer review history of their article (what does this mean?). If published, this will include your full peer review and any attached files.

Reviewer #1: No

Reviewer #2: No

Reviewer #3: No

---

## [Editor Report · Decision Letter 2]

10 Aug 2021

Dear Sarah,

We are pleased to inform you that your manuscript entitled "Endothelial Semaphorin 3fb regulates Vegf pathway-mediated angiogenic sprouting" has been editorially accepted for publication in PLOS Genetics. Congratulations!

Yours sincerely,

Cecilia Moens

Associate Editor

PLOS Genetics

Gregory P. Copenhaver

Editor-in-Chief

PLOS Genetics

Comments from the reviewers (if applicable):

**Data Deposition**

http://datadryad.org/submit?journalID=pgenetics&manu=PGENETICS-D-21-00136R2

**Press Queries**

---

## [Editor Report · Acceptance letter]

18 Aug 2021

PGENETICS-D-21-00136R2 

Endothelial Semaphorin 3fb regulates Vegf pathway-mediated angiogenic sprouting 

Dear Dr Childs, 

We are pleased to inform you that your manuscript entitled "Endothelial Semaphorin 3fb regulates Vegf pathway-mediated angiogenic sprouting" has been formally accepted for publication in PLOS Genetics! Your manuscript is now with our production department and you will be notified of the publication date in due course.

With kind regards,

Agnes Pap

PLOS Genetics

On behalf of:
